# Aerodynamic Effects of Gurney Flaps
# on the Rotor Blades of a Research Wind Turbine

Jörg Alber[1], Rodrigo Soto-Valle[1], Marinos Manolesos[2], Sirko Bartholomay[1], Christian Navid Nayeri[1], Marvin Schönlau[1], Christian Menzel[1], Christian Oliver Paschereit[1], Joachim Twele[3], Jens Fortmann[3]

[1] Technische Universität Berlin, Hermann-Föttinger Institut, Müller-Breslau-Str. 8, 10623 Berlin, Germany
[2] College of Engineering, Swansea University, Bay Campus, Fabian Way, Swansea, SA1 8EN, United Kingdom
[3] Hochschule für Technik und Wirtschaft Berlin, Wilhelminenhofstraße 75A, 12459 Berlin, Germany

*Correspondence to*: Jörg Alber (joerg.alber@htw-berlin.de)

**Abstract.**

This paper investigates the aerodynamic impact of Gurney flaps on a research wind turbine of the Hermann-Föttinger Institute at the Technische Universität Berlin. The rotor radius is 1.5 meters and the blade configurations consist of the clean and the tripped baseline cases emulating the effects of forced leading edge transition. The wind tunnel experiments include three operation points based on tip speed ratios of 3.0, 4.3 and 5.6, reaching Reynolds numbers of approximately $2.5 \cdot 10^5$. The measurements are taken by means of three different methods; Ultrasonic Anemometry in the wake, surface pressure taps in the mid-span blade region and strain gauges at the blade root. The retrofit applications consist of two Gurney flap heights of 0.5 % and 1.0 % in relation to the chord length, which are implemented perpendicular to the pressure side at the trailing edge. As a result, the Gurney flap configurations lead to performance improvements in terms of the axial wake velocities, the angles-of-attack and the lift coefficients. The enhancement of the root bending moments imply an increase of both the rotor torque and the thrust. Furthermore, the aerodynamic impact appears to be more pronounced in the tripped case compared to the clean case. Gurney flaps are considered a passive flow-control device worth investigating for the use on horizontal axis wind turbines.

## 1 Introduction

The energy yield of modern Horizontal Axis Wind Turbines (HAWTs) is supposed to be optimal while keeping the maintenance costs as low as possible over a lifetime of around 20 years. However, the performance of rotor blades faces serious challenges, two of which are early separation and roughness effects. Early separation is a problem especially in the inner blade region towards the root where the Angles-of-attack (AoA) are elevated due to structural constraints, such as limited chord-length and twist-angles, see Figure 1 (a). Over time, the resulting dynamic loads contribute to the material fatigue of the blade (Mueller-Vahl et al., 2012). For this reason, Passive Flow Control (PFC) devices, such as Vortex Generators (VGs), are implemented in the inner blade region of different-size HAWTs aiming at stall delay (Pechlivanoglou et al., 2013). At the

same time, the longstanding surface erosion causes roughness effects, especially close to the Leading Edge (LE), see Figure 1 (b). LE roughness is relevant throughout the entire blade span and especially in the outer region towards the blade tip. Apart from the broad range of weather conditions, surface roughening is aggravated by rain, insects as well as sand or salt particles (Pechlivanoglou et al., 2010). Consequently, the energy yield of HAWTs is often found lower than predicted or regressing over time (Wilcox et al., 2017).

(b)

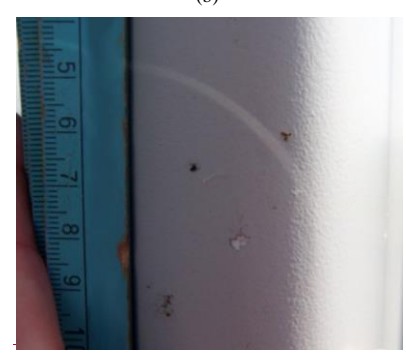

(a)

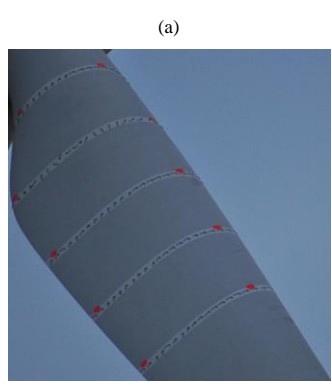

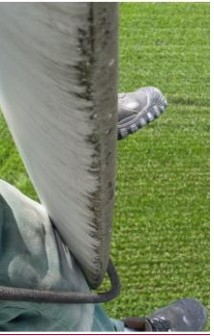

Figure 1. Rotor blades of utility scale wind turbines. (a) Flow indicators to detect early separation in the inner blade region, reproduced from Pechlivanoglou et al. (2013). (b) Leading edge ~~roughness~~erosion, with permission from Seilpartner Windkraft GmbH,~~reproduced from Pechlivanoglou et al. (2010).~~

This paper investigates the retrofit application of Gurney Flaps (GFs) in order to improve the aerodynamic performance of

rotor blades. This PFC device consists of a wedge- or right-angle profile that is attached perpendicular to the pressure side at

the Trailing Edge (TE). The GF-height, *GF*, in relation to the chord-length, *c*, is the main design parameter, illustrated in Figure 2Figure 2 (a). It is usually in the range of 0.5 %c < *GF* < 2.0 %c without taking the TE thickness into account.

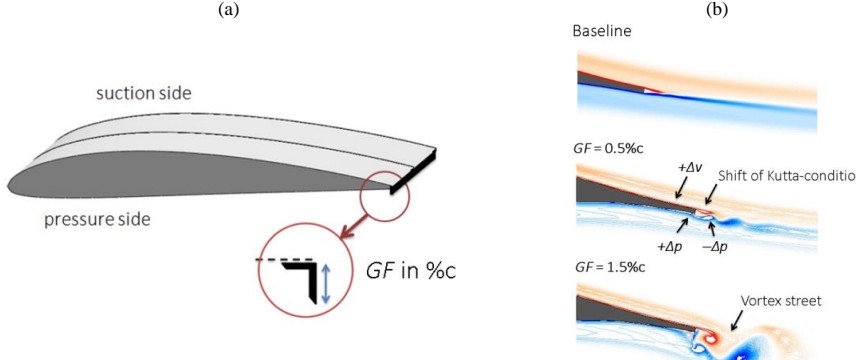

**Figure 2.** (a) Position of the Gurney flap at the trailing edge of a Clark-Y airfoil section. (b) CFD simulation of the HQ17 airfoil at $Re$ = $1.0 \cdot 10^6$, reproduced and modified from Schatz et al. (2004a).

The research on TE flaps of airplane wings dates back to the early 20th century (Gruschwitz and Schrenk, 1933). The GF itself is named after the racecar driver Dan Gurney who discovered the significant gain in downforce when applying the device on the rear spoilers. Following from that, GFs have been implemented on high-lift dependent transport airliners (Bechert et al., 2000) and helicopter stabilizers (Houghton et al., 2013). More recently, Vestas® has started offering GFs in combination with VGs as so-called aerodynamic upgrades of HAWTs, predicting annual yield improvements of up to 2.0 % (Vestas, 2020). The
design of the DTU 10 MW Reference Wind Turbine includes smooth wedge-shaped GFs in the first half of the blade length, 0.05R < *r* < 0.4R, using GF-heights in the range of 1.3 %c < *GF* < 3.5 %c (Bak et al., 2013).

Figure 2Figure 2 (b) illustrates the changes in the flow field of the laminar airfoil HQ17 when implementing different GF-heights, as reported by Liebeck (1978) by means of the Newman airfoil. Key to the aerodynamic understanding is the development of one vortex upstream and two counter-rotating vortices downstream of the GF, as such entailing a low-pressure
region in the TE wake. As a result, the downwash angle of the flow becomes steeper, the requirements for pressure recovery on the suction side milder, the local boundary layer thinner and the suction peak higher. Additionally, the flow on the pressure side decelerates leading to a positive pressure built-up in the TE region. The resulting shift of the Kutta-condition leads to increased circulation and thus to elevated lift forces, which is the main Gurney flap characteristic. At the same time, the low-pressure region aft the TE induces additional drag, especially if vortex shedding is initiated in the form of a Kármán vortex
street. Hence, the lift increase is accompanied by a certain drag penalty that affects the Lift-to-Drag (L/D) ratio accordingly.

This is why various experimental and numerical research projects aim at limiting the adverse drag increase while maintaining the beneficial lift enhancement. Giguère et al. (1995) and Kentfield (1996) conclude that the GF-height is supposed to be submerged into the local Boundary Layer (BL) in order to keep the drag on an acceptable level. Bechert et al. (2000) demonstrate that additional holes, slits and especially the pattern of dragonfly wings lead to reduced drag on the HQ17 airfoil ($th_{max}$ = 15.2 %c, $Re$ = 1.0·10$^6$). In addition, promising results are presented for very small GF-heights in the range of 0.2 %c < $GF$ < 0.5 %c, i.e. substantially smaller than the BL thickness at the TE. Following from that, CFD-based wake simulations of Schatz et al. (2004b) reveal that the amount of induced drag depends on the GF-height, in fact, in a disproportionate manner, as illustrated in Figure 2Figure 2 (b): for $GF$ = 1.5 %c a vortex street is triggered, while for $GF$ = 0.5 %c the wake is shed in a relatively smooth way. In a similar manner, Alber et al. (2017) suggest the use of very small GF-heights of approximately half the local BL thickness in order to maintain, or even improve, the airfoil L/D-ratio of different DU and NACA airfoils.

The aforementioned design principle, $GF$ < δ, is applied on the rotor blades of the Berlin Research Turbine (BeRT) using GF-heights of 0.5 %c and 1.0 %c. In addition, forced LE transition is triggered in order to emulate the effects of leading edge roughness.

The aerodynamic impact of GFs is investigated by means of the following measurement methods:

- 3D Ultrasonic Anemometry in the turbine wake to determine the local AoA.
- Chord-wise pressure taps to calculate the local pressure distribution and the lift performance.
- Strain gauges at the blade root to measure the flapwise and the edgewise root bending moments.

In summary, the objective of the experiments is to assess the suitability of retrofit GFs in order to alleviate the following adverse effects:

- Early separation due to the high AoA regime, relevant in the inner blade region, see Figure 1 (a).
- Decreasing lift forces due to leading edge erosion, relevant in the outer blade region, see Figure 1 (b).

In the remaining of this paper, the experimental set-up is described in detail, followed by the presentation and the discussion of the results. The main conclusions are summarized in the final section of this report.

## 2 Experimental set-up

### 2.1 Berlin Research Turbine

The BeRT is a test bench of the closed-loop wind tunnel of the Hermann-Föttinger Institut at the Technische Universität Berlin.
It is a unique wind turbine demonstrator to explore specific fluid-dynamic phenomena based on a fully equipped rotating system (Vey et al., 2015).

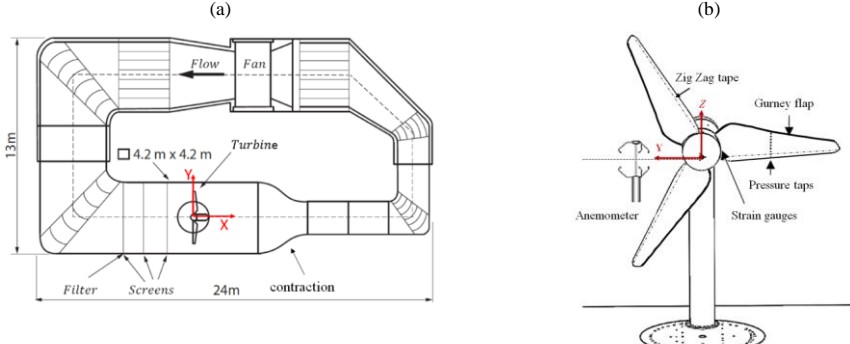

**Figure 3.** (a) Closed-loop wind tunnel in top-view, reproduced and modified from Klein at al. (2018). (b) BeRT set-up in front-view looking downstream.

Figure 3Figure 3 (a) depicts the wind tunnel facility consisting of the high speed (2.0 x 1.4) $m^2$ and the low speed (4.2 x 4.2)
$m^2$ test section. The BeRT is situated in the low speed test section downstream of the flow-conditioning screens and upstream of the wind tunnel contraction. The maximum inflow velocity is 10 ms$^{-1}$. The third screen upstream the rotor plane is equipped with an additional turbulence filter mat (Vildedon P15/150s) in order to reduce the turbulence intensity to 1.0 % $< Ti <$ 1.5 %, as reported by Bartholomay et al. (2017). Figure 3Figure 3 (b) displays the BeRT set-up and the measurement methods applied. The rotor radius is $R = 1.5$ m producing a relatively large blockage ratio of approximately 40 % in relation to the test section
area. The blockage effects on both the flow and the rotor performance are discussed in Sect. 3.1. Relative distances are expressed in relation to the rotor radius, $R$, and the zero position at the center of the rotor plane at $X = Y = Z = 0$. The blades consist of the low Reynolds-profile, Clark-Y, with a maximum thickness of $th_{max} = 11.9$ %c and a modified TE thickness of 0.75 %c. The blade geometry is optimized aerodynamically including a linear decrease of both the chord-lengths and the twist-angles from root to tip alongside most of the blade span. The root section is contiguous to the round rotor hub and the tip
section is pointy, see Figure 4Figure 4. The tip speed ratio at rated conditions is $TSR = 4.3$, developing a span-wise Re-number range from root to tip of $1.7 \cdot 10^5 < Re < 3.0 \cdot 10^5$. The axial inflow velocity is captured by two parallel Prandtl tubes that are permanently installed at approximately one rotor radius upstream, close to each wind tunnel wall and slightly above hub-height. At rated conditions, the inflow velocity is 6.5 ms$^{-1}$ at a rotational frequency of $f_{rot} = 3.0$ Hz. The data acquisition system

of the rotating sensors, such as pressure taps and strain gauges, is installed within the rotational spinner, see Figure 6~~Figure 6~~

(a). The electrical power is transferred to the rotating system through a slip ring. Communication with the host PC is established via WIFI connection in order to set and modify the rotational speed. The signals are captured on all channels simultaneously at a rate of 10 kHz generating around $6.0 \cdot 10^5$ data points per measurement which are streamed to a host PC via network connection.

## 2.2. Blade configurations and operation points

The test matrix consists of ~~four~~ six blade configurations (Table 1) and~~,~~ three operation points (Table 2), ~~and three measurement methods,~~ which are specified throughout this section.

**Table 1.** Blade configurations

|  | Tripped case | Clean case |
|---|---|---|
| Baseline |  |  |
| GF = 0.5 %c | Operation points |  |
| GF = 1.0 %c |  |  |

### 2.2.1 Forced transition

Following Klein et al. (2018), the principal baseline configuration of the BeRT includes Zig Zag (ZZ) turbulator tape, in short, the tripped case. ZZ tape is applied in order to initiate the laminar-to-turbulent transition of the Boundary Layer (BL) at a fixed location. In practical terms, it is used to emulate LE roughness effects on both airfoil sections (Rooij and Timmer, 2003) and rotor blades (Zhang et al., 2017). Its height is slightly smaller than the local BL thickness, $\delta$, in order to trigger the BL transition while avoiding a disproportionate drag increase or even turbulent separation. The ZZ tape is implemented on all BeRT blades

at a chord-wise LE position of both the Suction Side (SuS) at $x_{SuS} = 5.0$ %c and the Pressure Side (PrS) at $x_{PrS} = 10.0$ %c. The BL thickness of the clean baseline is calculated with the software XFOIL (Drela, 1989) based on the Re-number, the AoA and the N-criterion (Ncrit) modeling the transition location. The design conditions of the Clark-Y airfoil are defined by $\alpha_{opt} = 5.0°$, $Re = \approx 2.5 \cdot 10^5$ and Ncrit = 6 accounting for the elevated $Ti$ inside the test section (Sect. 2.1). As such, the attached flow at pre-stall conditions is assumed two-dimensional in order to estimate $\delta$ by means of the XFOIL code. The absolute height of the

ZZ tape is adjusted in various steps in relation to the chord-length, as depicted in Figure 4~~Figure 4~~ (a). In addition, all experiments are also performed under the consideration of the free BL transition, i.e. without including ZZ tape, in short, the clean case.

**Kommentiert [D2]:** In total six configurations were tested (rather than four). The reference to Table 1 was missing.

### 2.2.2 Gurney flaps

The GF-height is supposed to be submerged into the BL at the TE in order to keep the drag penalty on an acceptable level, as
discussed in Sect.1. Hence, it is important to estimate $\delta$ before dimensioning the GF-height, since the aerodynamic impact
depends on the $GF/\delta$ ratio. Apart from the AoA and the transition location, $\delta$ is related to $Re$. The Re-number range of the
BeRT is significantly lower compared to the blades of multi-MW HAWTs. At design conditions ($Re =\approx 2.5\cdot10^5$), the XFOIL
code predicts the BL thickness at the TE to be $\delta_{TE} = 1.0$ %c. Additionally, another GF-height of half the local $\delta$ is chosen, so
that the GF configurations consist of $GF = 1.0$ %c and $GF = 0.5$ %c. For comparison, the FFA-W3-241 airfoil ($th_{max} = 24.1$
%c, $Re = 12\cdot10^6$, $\alpha_{opt} =10.0°$), which is used in the outer blade region of the DTU 10MW Reference Wind TurbineRWT (Bak
et al., 2013), develops generates a BL of $\delta_{TE} \approx 0.30$ %c. As such, the application of $GF > 0.30$ %c would be likely to cause the
L/D ratio to decline, as illustrated in Figure 2Figure 2 (b).

Apart from the very tip section, the GFs are implemented in the form of thin angle profiles made of brass. One side of the
angle profiles is cut in a linear way in order to match the chord decrease, as shown in Figure 4Figure 4 (b). The other side of
the profile is attached with thin double-sided adhesive tape adjacent to the TE.

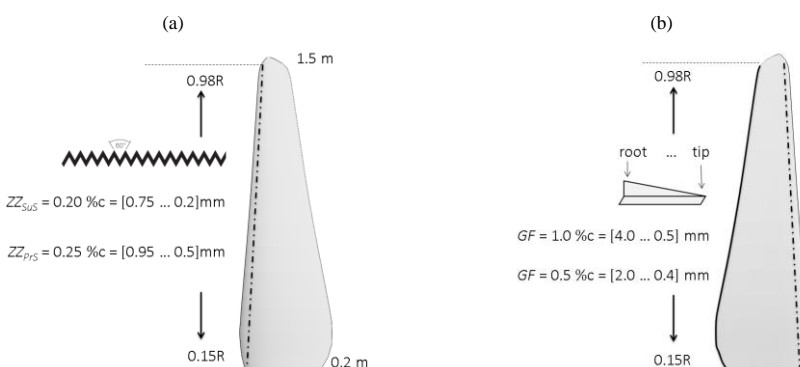

**Figure 4.** (a) Zig Zag tape at the leading edge of the suction side. (b) Gurney flap and ZZ tape at the pressure side of the trailing edge.

### 2.2.3 Operation points

The Operation Points (OPs) include the so-called stall, rated and feather conditions, which are characterized by low, medium
and high TSR or AoA, respectively, see Table 2. Each measurement has a total duration of 60 s. No blockage correction is
applied, so that the results refer to the conditions inside the closed test section. All sensors are calibrated and an zero-offset
measurement is performed before each test-run in order to reduce experimental errors. The uncertainty of the results is
evaluated in Appendix B.

**Table 2.** Summary of operation points.

|  | **Stall** | **Rated** | **Feather** |
|---|---|---|---|
| **TSR** | 3.0 | 4.3 | 5.6 |
| **Inflow velocity in m s$^{-1}$** | 6.5 | 6.5 | 5.0 |
| **Rot. frequency in Hz** | 2.1 | 3.0 | 3.0 |
| **Re-number (Sect. 3.2)** | $2.2 \cdot 10^5$ | $2.8 \cdot 10^5$ | $2.7 \cdot 10^5$ |
| **AoA in °<br>(tripped baseline, Sect. 3.1)** | 16.3 | 8.8 | 4.8 |
| **AoA in °<br>(clean baseline, App. A)** | 16.5 | 8.6 | 4.6 |

### 2.3 Measurement methods

The experimental approaches are summarized in Table 3 and explained in detail throughout this section.

**Table 3.** Measurement methods and quantities

| Sensor | Measured quantity | Derived quantity | Blade position |
|---|---|---|---|
| Ultrasonic Anemometer | 3D wake velocities | AoA | 0.56R |
| Pressure taps | Pressure distribution | Lift coefficients | 0.45R |
| Strain gauges | Flapwise and edgewise bending moments | | Blade root |

### 2.3.1 Ultrasonic anemometry

3D Ultrasonic Anemometers (UAs) are widely spread in the wind energy industry. The technology is recognized by different wind industry standards such as the IEC 61400 to determine the power curve of wind turbines or the Association of German Engineers (VDI) for turbulence measurements. There are numerous references for the use of UAs in the context of wind tunnel

campaigns, such as Weber et al. (1995), Hand et al. (2001) and Cuerva et al. (2003). The UA is a commercial product of Thies CLIMA (version 4.383). According to the manufacturer, they are pre-calibrated and free from maintenance.

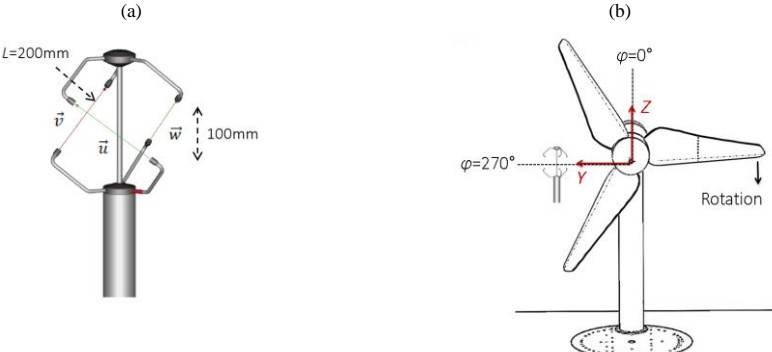

**Figure 5.** (a) Ultrasonic Anemometer, ~~reproduced and modified~~with permission from Thies CLIMA. (b) Definition of the azimuthal blade positions looking downstream.

Figure 5~~Figure 5~~ (a) displays the three separate acoustic transmitter-receiver pairs that are installed orthogonally to each other.

The velocity vectors, $\vec{u}$, $\vec{v}$ and $\vec{w}$, are determined by six individual measurements based on the bidirectional time-of-flight principle, i.e. the duration of each signal to be sent and received,

$$\vec{u} = \frac{L}{2}\left(\frac{1}{t_1} - \frac{1}{t_2}\right),$$  (1)

where $L$ is the exact running-length between each sensor pair, so that the measurement volume amounts to $(200 \ x \ 200 \ x \ 100)$ mm$^3$. The velocity vectors $\vec{v}$ and $\vec{w}$ are determined accordingly. Eq. (1)(1) shows that the 3D velocity calculation depends

solely on the average propagation time of the ultrasound, $t_1$ and $t_2$, depending on the specific airflow passing through the measurement volume. As such, the output values already imply the density and temperature of the air. Subsequently, the velocity vectors are transformed into a natural coordinate system, so that the output time-series consist of the axial, lateral and vertical velocity components, $u$, $v$ and $w$. The device-internal data acquisition system is a half-duplex interface that is completely independent of both the wind tunnel and the BeRT system. According to the manufacturer, the measurement

accuracy is 0.1 m s$^{-1}$ per integrated value and 0.01 m s$^{-1}$ with respect to each of the three velocity components. The data is recorded at a sampling rate of 60 Hz thus providing around 3600 data points per measurement. Considering the relatively large measurement volume and the low sampling rate compared to e.g. hotwire or laser-based devices, the UA is not adequate for the investigation of complex or high-speed flow structures. However, the BeRT wake-flow is expected to consist of an axial and a tangential velocity component due to the formation of a rotating wake tube. The impact of complex tip and root vortices

is considered negligible in the mid-span blade region, as shown by Herráez et al. (2018).

The UA is installed at one static position downstream, $X = 1.3R$, in the mid-span region, $Y = 0.56R$, and at hub height, $Z = 0R$, see Figure 5Figure 5 (b). It is positioned vertically with a spirit level and turned around its own axis towards the undisturbed axial inflow, so that the lateral and the vertical components, $v$ and $w$, tend to zero. The set-up is fixed at its final position for

all measurements, which are presented in Sect. 3.

**2.3.2 Pressure taps**

The pressure distribution is extracted by means of 18 Pressure Taps (PTs) on the SuS and 12 on the PrS, located along the chord-length at $r = 0.45R$, see Figure 6Figure 6 (b). Each orifice is connected via silicone tubing to its corresponding differential pressure sensor (HCL0025E), i.e. the pressure box inside the spinner. The sensor accuracy is given with 0.05 % of

the full scale range of $\pm 2500$ Pa under nominal conditions. The experimental procedure and the data post-processing is based on Soto-Valle et al. (2020).

(a)                                        (b)

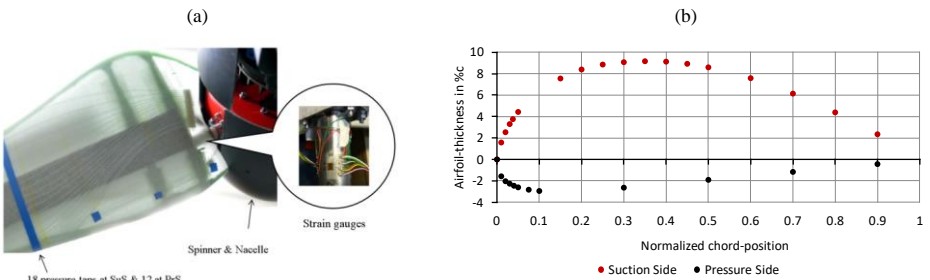

**Figure 6.** (a) BeRT blade and pressure taps, ~~reproduced and modified~~ with permission from SMART BLADE GmbH.~~Fischer (2015).~~ (b) Chord-wise position of pressure taps at $r = 0.45R$.

**Kommentiert [D3]:** The previous reference is abundant as it refers to an internal document. After talking back to the company, Fischer (2015) was replaced by SMART BLADE GmbH.

The differential pressure values are transformed into the pressure coefficient,


$$c_{pi} = \frac{\Delta p_{sti} + p_{rot}}{p_{dyn,ref}} = \frac{(p_{sti} - p_{st,\infty}) + (0.5\rho \cdot (\omega r)^2)}{p_{dyn,ref}},$$ (2)

where

- $\Delta p_{sti}$ is the static pressure difference between each PT and the inflow Prandtl tube $p_{st,\infty}$.
- $p_{rot}$ refers to the pressure due to the rotation of the blade element. It is added to $\Delta p_{sti}$ in the form of a constant correction term in accordance with Hand et al. (2001).

- $p_{dyn,ref}$ describes the referential dynamic pressure, i.e. the effective flow velocity experienced by the blade element. Following Hand et al. (2001), it is determined by the maximum pressure that is recorded on the pressure side, the frontal stagnation point, where $c_{pi} = 1.0$. According to Eq. (2)(2) the referential dynamic pressure is then calculated with $p_{dyn,ref} = \Delta p_{st,ref} + p_{rot}$.

The $c_p$ values are phase-averaged over an azimuthal angle of $\varphi = 10°$, see Figure 5Figure 5 (b). Each PT provides a total of 36 pressure values at the following blade positions: $\varphi = [0°, 10°, 20° ... 350°]$, so that $\varphi = 270°$ contains the average of all data points between $265° < \varphi < 275°$. The pressure difference, $\Delta c_p$, is calculated by subtracting the integrated $c_p$ distribution between the PrS and the SuS in order to determine both the normal coefficient, $c_n$, and the tangential coefficient, $c_t$. Per definition, $\vec{c_n}$ is orthogonal to the chord-line pointing towards the SuS, while $\vec{c_t}$ is parallel to the chord-line pointing towards the LE.


According to Hand et al. (2001), the axial and tangential coefficients are calculated with

$$c_n = \frac{1}{2} \cdot \sum_{i=1}^{30} (c_{pi} + c_{pi+1}) \cdot (x_{i+1} - x_i)$$ (3)

and

$$c_t = \frac{1}{2} \cdot \sum_{i=1}^{30} (c_{pi} + c_{pi+1}) \cdot (y_{i+1} - y_i),$$ (4)

where $x$ and $y$ are the normalized chord positions of each PT. The numbering starts at the TE ($x = 0.9$) with the PTs on the SuS, moving anti-clock wise until the LE ($x = 0$) and back to the TE on the PrS.

Subsequently, the lift coefficient, $c_l$, and the pressure drag coefficient, $c_{dp}$, are determined by (Fuglsang et al., 1998)

$$c_l = c_n \cdot \cos(\alpha) + c_t \cdot \sin(\alpha)$$ (5)

and

$$c_{dp} = c_n \cdot \sin(\alpha) - c_t \cdot \cos(\alpha).$$ (6)

The required AoA, α, are adopted by the uncorrected inflow and wake velocity measurements (Sect. 3.1). At pre-stall conditions, i.e. considering small AoA, $c_t \ll c_n$, so that $c_n \approx c_l$ (Barlow et al., 1999). It is noted that Eq. (6)(6) describes the pressure drag which does not account for the skin-friction drag component. Hence, it is not possible to extract the total drag, $c_d$, of the blade element via the local $c_p$ distribution (Houghton et al., 2013).

### 2.3.3 Strain gauges

The Strain Gauges (SGs) are mounted at the clamping of the blade detecting the Root Bending Moments (RBMs) in the out-of-plane or flapwise and in-plane or edgewise direction, see Figure 6Figure 6 (a). They are connected in a full-bridge configuration aiming at the mitigation of temperature and cross talk effects (FAET-A6194N-35). The experimental procedure to determine the RBMs is based on Bartholomay et al. (2018). For the purpose of the presented baseline measurements, a simplified post-processing protocol is applied without including the data-based cross talk correction.

Before testing each blade configuration, the offset signal is recorded in slow-motion at the lowest rotating frequency available, $f_{rot} = 0.1$ Hz. In this way, the gravitational RBMs are subtracted from the results, which are otherwise registered as a sinusoidal signal in the edgewise direction. At operational frequencies, the axial forces due to the blade rotation are causing a material deformation directed towards the blade tip. They are quantified as a combination of centrifugal and gravitational forces by

$$F_{axial} = F_{cent} - F_{grav} = \left(m_{blade} \cdot r_{cg} \cdot \omega^2\right) - \left(m_{blade} \cdot g \cdot \cos(\varphi)\right), \tag{7}$$

where $m_{blade} = 5.67$ kg, the center of gravity is located at $r_{cg} = 0.31R$, $g$ is the gravitational constant and $\varphi$ refers to each phase-locked blade position. The rotational frequency, $\omega$, is kept constant during each test-run so that the centrifugal force $F_{cent}$ becomes a constant correction term at each OP. The effective flapwise and edgewise RBMs, which are related exclusively to the aerodynamic loads acting on the blade, are determined by

$$M_{flap}(\varphi) = \left(U_{f,\text{raw}}(\varphi) - U_{f,off}(\varphi)\right) \cdot K_{f1} - \left(F_{axial} \cdot K_{f2}\right) \tag{8}$$

and

$$M_{edge}(\varphi) = \left(U_{e,\text{raw}}(\varphi) - U_{e,off}(\varphi)\right) \cdot K_{e1} - \left(F_{axial} \cdot K_{e2}\right), \tag{9}$$

where

- $M_{flap}$ and $M_{edge}$ are the aerodynamic flapwise or edgewise RBMs in Nm.
- $U_{f,raw}$ and $U_{e,raw}$ stand for the raw data signal in V.
- $U_{f,off}$ and $U_{e,off}$ describe the slow-motion offset signal in V.
- $K_{f1}$ and $K_{e1}$ refer to constant calibration factors to transform V into Nm.

• $K_{f2}$ and $K_{e2}$ refer to constant calibration factors to transform the axial forces from N into Nm.

Applying Eq. (8)(8) and (9) both the out-of-plane and the in-plane RBMs are computed for each of the 36 blade positions, see Sect. 3.

## 3 Results

The measurement results of both the tripped and the clean cases are presented and discussed. For space economy, the clean case is only included presented in terms of the concluding results, such as the lift performance in Sect. 3.2 and the root bending moments in Sect. 3.3, but otherwise accessible in Appendix A for completeness.

### 3.1 Wake velocities and angles-of-attack

Following Snel et al. (2009), Figure 7(a) shows the average axial and tangential wake velocity normalized by the axial inflow
velocity at each OP, $uu_\infty^{-1}$ and $wu_\infty^{-1}$.

(a)                                                                              (b)

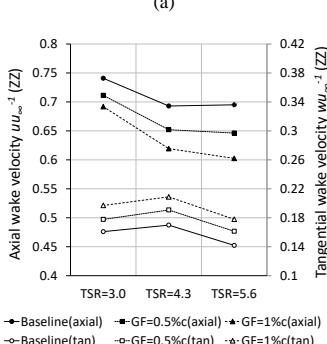
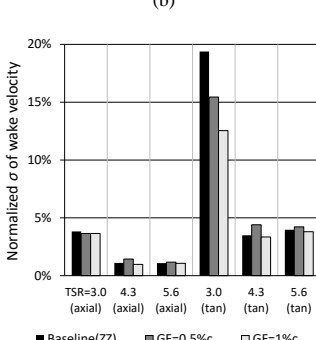

**Figure 7.** Tripped case at $r = 0.56R$ and $\varphi = 270°$. (a) Mean axial and tangential (tan) wake velocity normalized by the inflow velocity. (b) Standard deviation of the wake velocity normalized by the average wake velocity.

Starting from the baseline, Figure 7Figure 7 (a) shows that the axial wake velocities are found to be significantly higher compared to typical free flow conditions without wind tunnel walls. According to the steady state Blade Element Momentum
(BEM) method, the optimum axial wake velocity is supposed to be around one third of the inflow (Burton et al., 2011). In this case, it amounts to more than two thirds at all OPs. This phenomenon is caused by the wind tunnel blockage effects, previously shown by CFD simulations using the fluid dynamic code FLOWer. At rated conditions of the BeRT, Klein et al. (2018) conclude that the flow decelerates to an axial wake velocity in the range of $0.62u_\infty < u_{CFD} < 0.77u_\infty$, which is in agreement with the experimental results, $u_{exp} = 0.69u_\infty$. The corresponding tangential velocity, on the other hand, is similar to the steady state

BEM simulation of QBlade (Marten et al., 2013) with $w_{BEM} = 0.18u_\infty$ compared to $w_{exp} = 0.17u_\infty$. According to Eq. (11), $w$ depends primarily on the rotational speed of the blade. ~~As such, t~~The tangential wake velocity is therefore less affected by the wind tunnel blockage effect.

Regarding the impact of the GFs, Figure 7~~Figure 7~~ (a) illustrates the consistent decrease of the axial, and the consistent increase of the tangential wake velocity in relation to the GF-height. The lateral velocity component is neglected as it amounts to $v \ll |\ 0.1\ ms^{-1}|$. Figure 7~~Figure 7~~ (b) shows the standard deviation normalized by the corresponding average velocity component ~~, as such~~ describing the 1D turbulence intensity, expressed in percent (Burton et al., 2011). As expected, the flow separation, $TSR = 3.0$, is captured by the UA in the form of a more turbulent wake field, especially regarding the tangential component. The GF configurations do not influence the wake turbulence considerably, except for the tangential velocity component at stall, where the GFs appear to mitigate the turbulence level.

According to the BEM method (Hansen, 2015), the wake velocity is converted into the axial and tangential rotor induction factors,

$$a = \frac{1}{2}\left(1 - \frac{u}{u_\infty}\right) \tag{10}$$

And

$$a' = \frac{w}{2\omega r}. \tag{11}$$

The induction factors, $a$ and $a'$, describe the decrease of the axial, and the increase of the tangential velocity component from a reference point sufficiently far away from the rotor plane rather than the rotor plane itself (Burton et al., 2011). The wake measurements are recorded at a distance of $X = 1.3R$ downstream in order to avoid the influence of the wind tunnel contraction, see Figure 3~~Figure 3~~ (a).

Subsequently, the AoA are derived by means of Eq. (10)~~(10)~~ and (11) with

$$\alpha = \arctan\left(\frac{(1-a)\ u_\infty}{(1+a')\ \omega r}\right) - \beta = \arctan\left(\frac{u_\infty + u}{2\omega r + w}\right) - \beta, \tag{12}$$

where the twist-angle at the radial location of the UA is $\beta\ (0.56R) = 9.8°$.

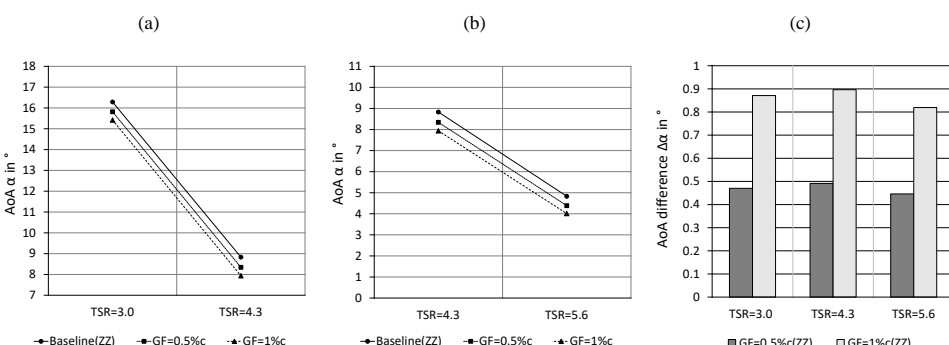

(a)                              (b)                              (c)

**Figure 8.** Angles-of-attack in the tripped case at $r = 0.56R$ and $\varphi = 270°$. (a) Stall and rated conditions (b) Rated and feather conditions (c) AoA difference between Gurney flap configurations and the baseline.

At rated conditions, the AoA of the baseline case is $\alpha_{ZZ} = 8.8°$, see ~~Figure 8~~Figure 8 (a) and (b). This outcome is in ~~general~~ agreement with different experimental and numerical investigations of the BeRT, gathered in Table 4.

**Table 4.** Comparison of approximate AoA results at rated conditions and $\varphi = 270°$.

| Method | Blade position | Case | AoA | Reference |
|--------|----------------|------|-----|-----------|
| Pressure taps | 0.45R | clean | 8.0 ° | Soto-Valle et al. (2020) |
| Ultrasonic Anemometry | 0.56R | tripped | 8.8 ° | Present study |
| 3-hole probe | 0.65R | tripped | 8.5 ° | Klein et al. (2018) |
| CFD simulation | 0.65R | tripped | 8.2 ° | Klein et al. (2018) |

~~The different experiments (Table 4) result in local AoA that are significantly higher compared to the original blade design of~~
~~the BeRT, $\alpha_{opt} = 5.0°$. Due to the built-in twist angles, the AoA is considered constant in the mid-span region, i.e. within the~~
~~range of $0.45R \leq r \leq 0.65R$.~~ The relatively small deviations between the results are due to the different measurement methods as well as blade configurations (Table 4). The AoA is therefore considered constant in the mid-span region within the range of $0.45R \leq r \leq 0.65R$. In all cases, the AoA are significantly higher compared to the original blade design of the BeRT, $\alpha_{opt} = 5.0°$.

> **Kommentiert [D4]:** This statement was corrected without changing the content.

~~Next,~~ Figure 8~~Figure 8~~ (c) displays the consistent AoA decrease caused by the GF configurations. The AoA differences between GF and Baseline configurations amount to $\Delta\alpha_{GF=0.5\%c} = 0.5°$ and $\Delta\alpha_{GF=1.0\%c} = 0.9°$, i.e. to a level that is closer to the

optimum blade operation. ~~As such, t~~The results quantify an important effect of retrofit GFs on the blade performance; decreasing axial wake velocities and thus reduced AoA.

In the following Sect. 3.2, the AoA are correlated with the normal force coefficients in order to obtain the lift coefficients.

### 3.2 Pressure distribution and lift performance

Figure 9~~Figure 9~~ shows the distribution of the pressure coefficients, $c_p$, in relation to the different OPs.

(a)               (b)               (c)

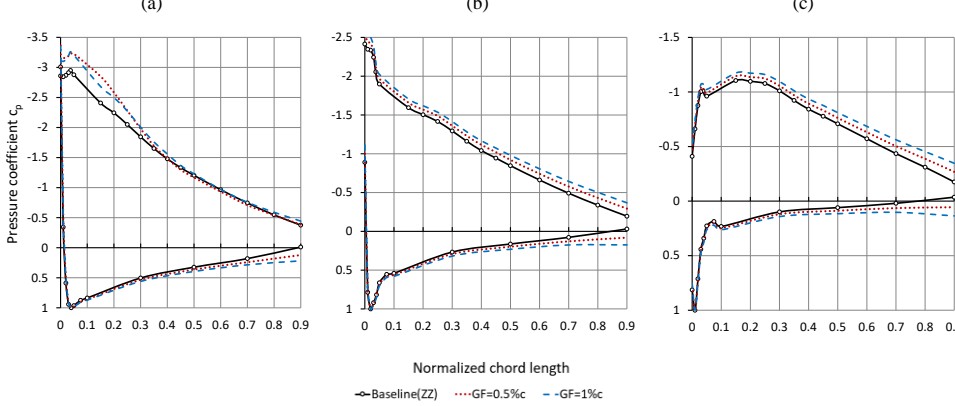

**Figure 9~~10~~.** Pressure coefficients in the tripped case with respect to different scales at $r = 0.45R$ and $\varphi = 270°$. (a) $TSR = 3.0$. (b) $TSR = 4.3$. (c) $TSR = 5.6$.

The $c_p$ curves shown in Figure 9~~Figure 10~~ (b) and (c) represent the pre-stall cases at $\alpha_{TSR=4.3} = 8.8°$ and $\alpha_{TSR=5.6} = 4.8°$, respectively. At stall, see Figure 9~~Figure 10~~ (a), the separation at the SuS is not yet complete despite the elevated AoA, $\alpha_{TSR=3.0} = 16.3°$. The curves indicate the effect of stall delay due to the blade rotation, as discussed hereafter.

~~Moreover, T~~the GF configurations cause an expansion of the pressure differences between the PrS and the SuS, $\Delta c_p$, along the complete chord-length and regarding all OPs. This effect is particularly visible in terms of the aft-loading towards the TE at $0.5 < x < 0.9$. ~~T~~ ~~As such, $\Delta c_p$ reflects~~ the increased circulation due to the GF applications is reflected by $\Delta c_p$, as reported by Storms and Jang (1994) based on the clean NACA 4412 airfoil ($th_{max} = 12.0 \%c$, Re $= 2.0·10^6$).

In order to quantify the results, the $c_p$ distribution is transformed into the local lift curve based on Eq. (5~~5~~). The required AoA are adopted from Sect. 3.1, so that the lift coefficients combine the results of both the wake velocity and the pressure measurements.

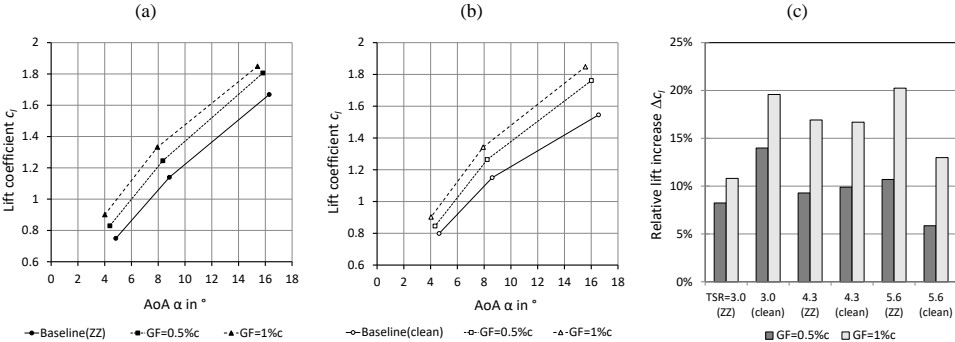

**Figure 10~~11~~.** Lift coefficients over angles-of-attack at $r = 0.45R$ and $\varphi = 270°$. (a) Tripped case. (b) Clean case. (c) Relative lift increase of Gurney flap configurations in relation to the corresponding baseline.

Figure 10~~Figure 11~~ (a) and (b) depict the lift coefficients of both the tripped and the clean cases. Starting from the baseline, the tripped case shows smaller $c_l$ at $4° < \alpha < 5°$ because of the forced BL transition at the LE. At $8° < \alpha < 9°$, this is not the case anymore, while in the stall region, $15° < \alpha < 17°$, the ZZ tape appears to develop a beneficial effect on the lift performance.

This phenomenon is probably caused by the tripped and more turbulent BL that remains attached until closer to the TE. In the clean case, however, the less energetic BL separates earlier thus leading to smaller $c_l$ at elevated AoA. This observation is confirmed by comparable airfoil experiments on the FX 63-137 airfoil section ($th_{max} = 13.7 \%c$, Re = $2.0 \cdot 10^5$) using ZZ tape with a thickness of 0.75 mm (Holst et al., 2016). Despite the decrease in the pre-stall, the lift coefficients are found on a similar level in the post-stall region.

Looking at the GF configurations, the $c_l$ performance in the tripped case is on a similar, or even higher level considering the complete AoA range, $4° < \alpha < 17°$. Hence, forced LE transition does not neutralize or mitigate the GF effect. In fact, the GF configurations appear to alleviate the adverse effects of _forced_ LE ~~roughness~~ transition by improving the local $c_l$ performance. Figure 10~~Figure 11~~ (c) highlights the relative lift increase, $\Delta c_l$, between the GF and the corresponding baseline configurations.

At rated conditions, $TSR = 4.3$, $\Delta c_{l,GF=0.5\%c} = 0.11$ or 9.3 % and $\Delta c_{l,GF=1.0\%c} = 0.19$ or 16.9 %, illustrating the main characteristic of retrofit GFs; the considerable lift increase.

The level of both $c_{l,baseline}$ and $\Delta c_{l,GF=1.0\%c}$ is in agreement with comparable wind tunnel experiments based on a similar Clark-Y airfoil section, as depicted in Figure 11~~Figure 12~~.

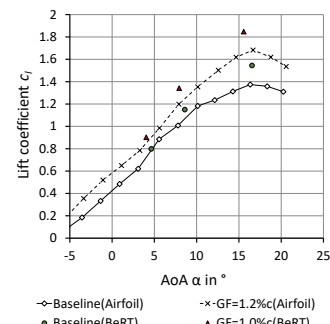

**Figure 11~~12~~.** Lift coefficients of the Clark-Y airfoil including Gurney flap, reproduced and modified from Kheir-Aldeen (2014).

Figure 11~~Figure 12~~ compares the lift coefficients of the clean Clark-Y airfoil section ($th_{max}$ = 14.0 %c, $Re$ = 2.1·10$^5$, $GF$ = 1.2 %c) and the clean Clark-Y blade element of the BeRT ($th_{max}$ = 11.9 %c, $Re$ = 2.5·10$^5$, $GF$ = 1.0 %c). The results demonstrate similarities for both the baseline and the GF configurations. The elevated $c_l$ in case of the BeRT are due to the thinner Clark-
Y blade element. At $c_{l,max}$, the blade performance is furthermore characterized by the radial flow due to the blade rotation causing stall delay. This behavior is in agreement with experiments on the field rotor at the Delft University of Technology. Rooij and Timmer (2003) report a significant shift of $c_{l,max}$ compared to 2D airfoil simulations.

For completeness, the lift over the pressure drag coefficients (Eq. (6~~6~~)) are displayed as an indicator of the drag performance.
It is reiterated that $c_{dp} < c_d$, as previously discussed in Sect. 2.3.2.

**Figure 12~~13~~.** Lift over pressure drag coefficients at $r$ = 0.45R and $\varphi$ = 270°. (a) Tripped case. (b) Clean case. (c) Pressure drag coefficients in relation to the corresponding baseline.

Figure 12Figure 13 (a) and (b) illustrate the dependency of $c_{dp}$ on the OP, reaching values of $0.024 < c_{dp,pre\text{-}stall} < 0.04$ and
$c_{dp,stall} \approx 0.25$. In general, the baseline results are comparable to the clean S809 airfoil ($th_{max} = 21.0\,\%c$, Re $= 3.0 \cdot 10^5$) that is
used for the NREL Phase VI test turbine (Hand et al., 2001). Figure 12Figure 13 (c) visualizes the increase of $c_{dp}$ in the tripped
case due to the implementation of the ZZ tape. The GF configurations, on the other hand, influence the $c_{dp}$ values in a less
noticeable way.

After evaluating one area of the mid-span blade region, the impact of GFs over the complete blade span is presented in Sect.
3.3

### 3.3 Root bending moments

The integration of the aerodynamic loads, i.e. the lift and the drag forces acting along the blade span, yield the RBMs. The in-
plane or edgewise RBMs are proportional to the rotor torque and thus the mechanical power output. They are directly related
to the out-of-plane or flapwise RBMs, which are proportional to the rotor thrust and thus the structural loads (Hansen, 2015).

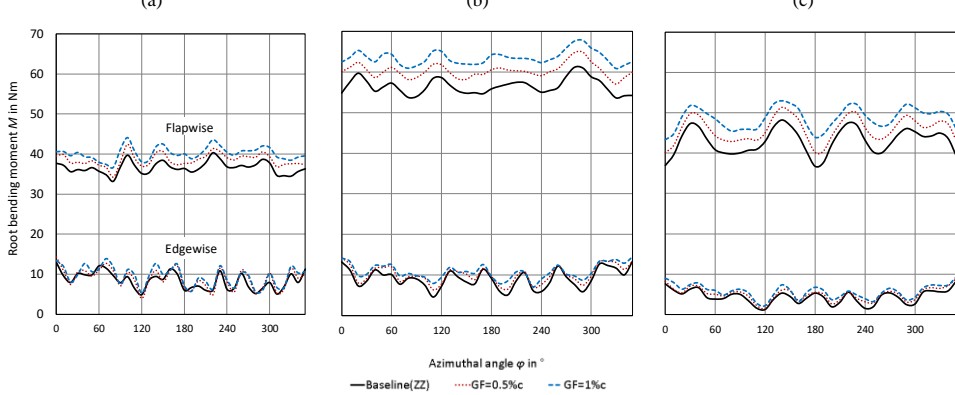

**Figure 1314.** Flapwise and edgewise root bending moments in the tripped case. (a) $TSR = 3.0$. (b) $TSR = 4.3$. (c) $TSR = 5.6$.

Figure 13Figure 14 displays the aerodynamic RBMs that are recorded over one blade revolution in the form of 36 phase-locked
blade positions. The impact of the GF configurations is registered as an overall increase of both the flapwise and the edgewise
RBMs. In order to quantify and to discuss the results, the RBMs are presented as average values for both the tripped and the
clean cases.

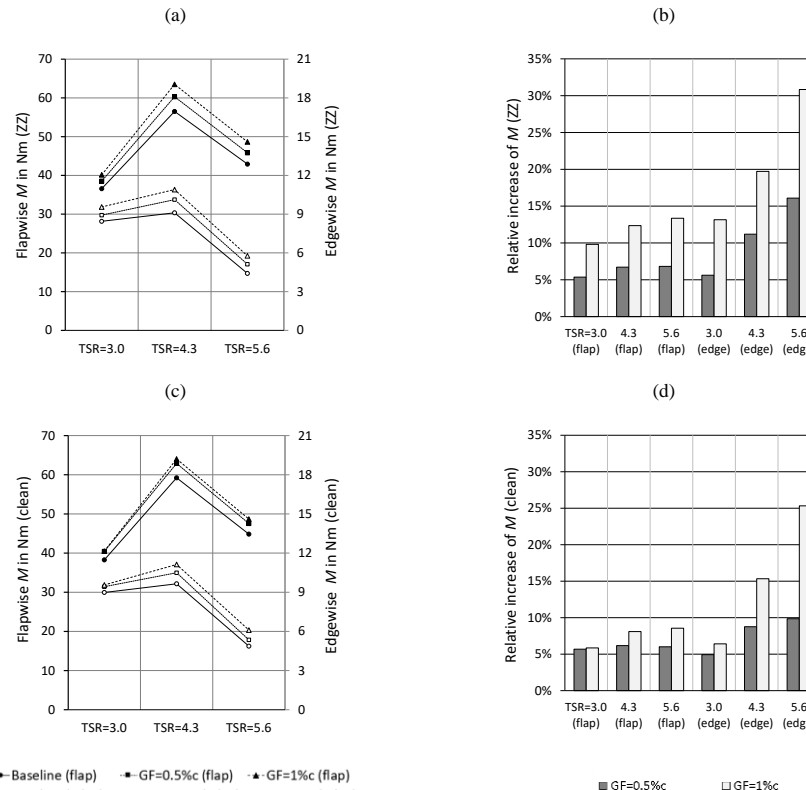

**Figure 1415.** Flapwise (flap) and edgewise (edge) root bending moments. (a) Tripped case. (b) Relative increase to tripped baseline. (c) Clean case. (d) Relative increase to clean baseline.

The results of Figure 14Figure 15 (a) confirm the increment of the average RBMs in relation to the GF-height in accordance with the previous Figure 13Figure 14. In the clean case, the overall trend is similar to the tripped case considering all OPs, see

Figure 14Figure 15 (c). This means that the impact of the Gurney flaps, previously quantified in terms of the local lift coefficients, is now registered in the form of increased RBMs in both the flapwise and the edgewise direction.

In Figure 14Figure 15 (b), the performance of the GF configurations is quantified in relation to the tripped baseline. At rated conditions, the average increase of the flapwise RBMs amount to $\Delta M_{\text{flap,GF}=0.5\%c} = 3.8$ Nm or 6.7 % and to $\Delta M_{\text{flap,GF}=1.0\%c} = 7.0$

Nm or 12.4 %. At the same time, the edgewise RBMs are enhanced by $\Delta M_{\text{edge,GF=0.5\%c}}$ = 1.0 Nm or 11.2 % and $\Delta M_{\text{edge,GF=1.0\%c}}$ = 1.8 Nm or 19.7 %. In the clean case, see Figure 14Figure 15 (d), the overall trend is similar, however less pronounced. In both cases, the GF configurations generate performance improvements regarding the rotor torque, albeit at the expense of the inherent increase of the rotor thrust.

Overall, the results reinforce the observation that GFs are more effective in relation to the tripped compared to the clean baseline. Looking at the relative increase shown in Figure 14Figure 15 (b) and (d), the GF configurations appear to alleviate the effects of forced LE transition, especially on the edgewise RBMs, as previously discussed in Sect. 3.2 with respect to the local lift performance.

**4 Conclusions**

The aerodynamic impact of Gurney flaps is investigated on the rotor blades of the Berlin Research Turbine. The test matrix consists of four blade configurations including the clean and the tripped baseline cases, as well as two GF configurations of 0.5%c and 1.0 %c. Furthermore, tThree measurement methods are applied, including 3D Ultrasonic Anemometry, surface pressure taps and strain gauges.

The baseline measurements confirm the influence of the prevailing wind tunnel blockage. At rated conditions, $TSR$ = 4.3, and in the mid-span blade region, the axial wake velocity is approximately double in comparison to ideal free flow conditions, i.e. without wind tunnel walls. As such, tThe corresponding angles-of-attack is elevated in comparison to the optimum blade design and amounts to $\alpha_{exp}$ = 8.8° rather than $\alpha_{opt}$ = 5.0°.

The impact of the Gurney flaps is registered regarding all blade configurations and operation points. In the tripped case and at rated conditions, the axial wake velocities are reduced and the angles-of-attack are decreased by $\Delta\alpha_{GF=0.5\%c}$ = 0.5° and $\Delta\alpha_{GF=1.0\%c}$ = 0.9°. At the same time, the local lift coefficients are enhanced by $\Delta c_{l,GF=0.5\%c}$ = 0.11 or 9.3 % and $\Delta c_{l,GF=1.0\%c}$ = 0.19 or 16.9 %, which is the main characteristic of Gurney flaps. The effect of the aerodynamic loads over the complete blade span is analyzed by means of the root bending moments. The average increase in the out-of-plane direction amounts to $\Delta M_{\text{flap,GF=0.5\%c}}$

= 3.8 Nm or 6.7 % and to $\Delta M_{\text{flap,GF=1.0\%c}}$ = 7.0 Nm or 12.4 %. Simultaneously, the in-plane bending moments are elevated by $\Delta M_{\text{edge,GF=0.5\%c}}$ = 1.0 Nm or 11.2 % and $\Delta M_{\text{edge,GF=1.0\%c}}$ = 1.8 Nm or 19.7 %. Hence, decreasing angles-of-attack and increasing lift coefficients appear to be correlated with the enhancement of both the rotor torque and the thrust. Overall, the aerodynamic effect is found more pronounced in the tripped case compared to the clean case.

The experimental results demonstrate the potential of retrofit Gurney flaps to improve the rotor blade performance in the following ways:

- Decreasing angles-of-attack to a level that is closer to the optimum blade operation.
- Elevated lift forces compensating for the adverse effects of forced leading edge transition.


In summary, Gurney flaps are considered a passive flow-control device worth investigating for the use on horizontal axis wind turbines of different sizes. However, the design of the Gurney flap-height in relation to the local boundary layer thickness is crucial in order to achieve performance improvements while avoiding detrimental effects such as additional drag forces. Future research is required to quantify the impact of Gurney flaps on dynamic loads, surface roughness and the power output of rotor

blades that operate in open field conditions and at high Reynolds numbers.

**Appendix A: Results of the clean case**

(a)                                                        (b)

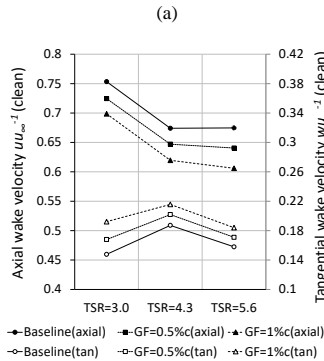

**Figure A 1.** Clean case at $r = 0.56R$ and $\varphi = 270°$. (a) Axial and tangential (tan) wake velocity normalized by the inflow velocity. (b) Standard deviation of the wake velocity normalized by the average wake velocity.


(a)                              (b)                              (c)

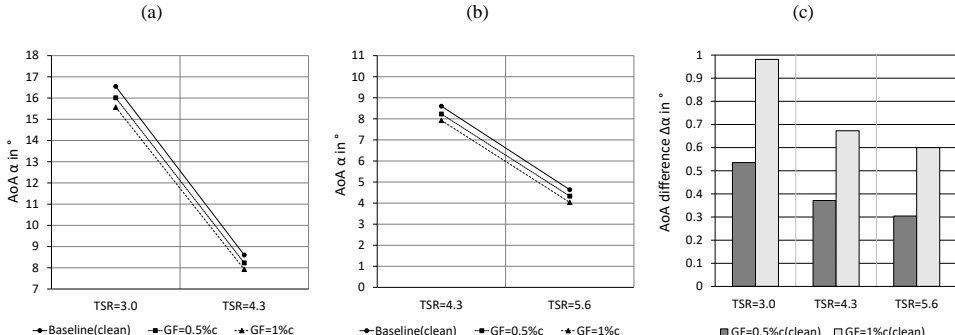

**Figure A 2.** Angles-of-attack in the clean case at $r = 0.56R$ and $\varphi = 270°$. (a) Stall and rated conditions (b) Rated and feather conditions (c) AoA difference between Gurney flap configuration and the baseline.

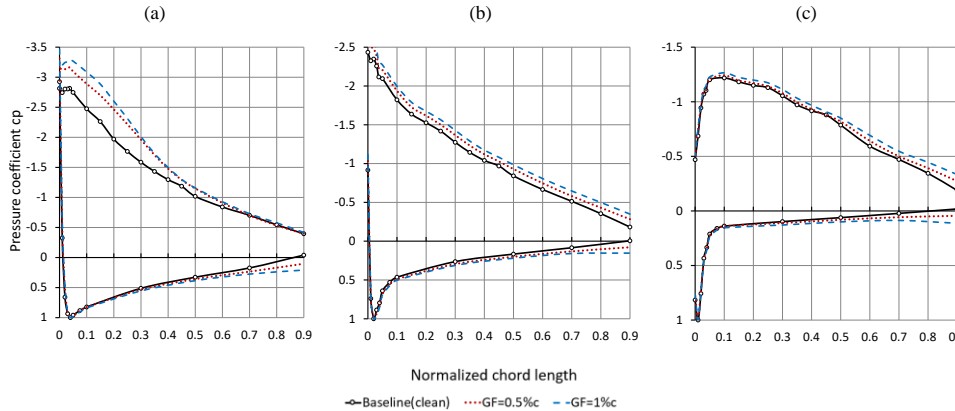

**Figure A 3.** Pressure coefficients in the clean case with respect to different scales at $r = 0.45R$ and $\varphi = 270°$. (a) $TSR = 3.0$. (b) $TSR = 4.3$. (c) $TSR = 5.6$.

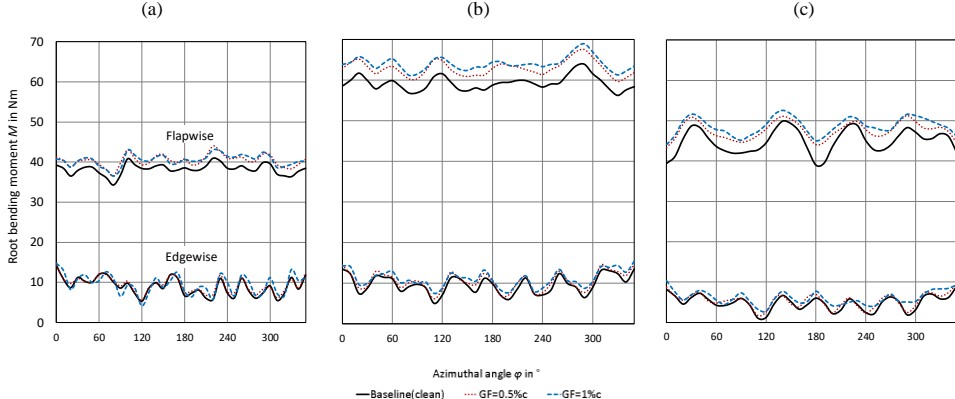

**Figure A 4.** Flapwise and edgewise root bending moments in the clean case. (a) $TSR = 3.0$. (b) $TSR = 4.3$. (c) $TSR = 5.6$.

## Appendix B: Uncertainty estimation

The experimental uncertainty of the raw results is expressed by means of the standard deviation,

$$\sigma = \sqrt{\frac{1}{n-1}\sum_{i=1}^{n}|\mu_i - \bar{\mu}|^2},$$

(13)

where $n$ is the number of samples and $\bar{\mu}$ refers to the average result. The values of $\sigma$ are rounded up conservatively and are considered ~~thus~~ representative for both tripped and clean baseline cases as well as the GF configurations.

**Table 5.** Standard deviation and reference values in brackets.

| Section | Quantity | *TSR* = 3.0 | *TSR* = 4.3 | *TSR* = 5.6 |
|---|---|---|---|---|
| 3.1 | $\sigma (u_\infty)$ [m s$^{-1}$] | 0.02 (6.57) | 0.02 (6.57) | 0.01 (5.02) |
| | $\sigma (u)$ [m s$^{-1}$] | 0.20 (4.87) | 0.06 (4.55) | 0.04 (3.49) |
| | $\sigma (w)$ [m s$^{-1}$] | 0.20 (1.06) | 0.06 (1.12) | 0.03 (0.71) |
| 3.2[a] | $\sigma_{min} (\Delta p)$ [Pa] | 2.8 (21.8) | 2.6 (102.5) | 1.7 (6.1) |
| | $\sigma_{max} (\Delta p)$ [Pa] | 30.0 (-193.6) | 5.8 (-269.1) | 3.2 (-41.6) |
| 3.3 | $\sigma (M_{flap})$ [Nm] | 1.9 (36.6) | 2.9 (56.5) | 2.2 (42.9) |
| | $\sigma (M_{edge})$ [Nm] | 1.0 (8.5) | 1.1 (9.1) | 0.6 (4.4) |

[a] Minimum and maximum standard deviation of pressure taps

As expected, the scatter of both the velocity and the pressure data depends on the OP, i.e. it is higher at stall (*TSR* = 3.0), see Table 5~~Table 5~~. Looking at the RBMs, however, the experimental uncertainty of $\sigma (M_{flap})$ and $\sigma (M_{edge})$ is influenced by the structural impact of the rotational frequency that the SGs register simultaneously to the aerodynamic forces. Overall, the standard deviation is not significantly influenced by either of the GF configurations.

Subsequently, the 95% confidence interval or so-called random error is computed with

$$\varepsilon = t \cdot \frac{\sigma}{\sqrt{n}} \approx 1.96 \cdot \frac{\sigma}{\sqrt{n}}$$

(14)

where $t$ is the Student's t-distribution (Barlow et al., 1999).

**Table 6.** 95% confidence interval and reference values in brackets.

| Section | Quantity | $TSR = 3.0$ | $TSR = 4.3$ | $TSR = 5.6$ |
|---|---|---|---|---|
| | $\varepsilon\,(u_\infty)$ [ms$^{-1}$] | $5.0 \cdot 10^{-5}$ (6.57) | $5.0 \cdot 10^{-5}$ (6.57) | $2.8 \cdot 10^{-5}$ (5.02) |
| 3.1 | $\varepsilon\,(u)$ [ms$^{-1}$] | $6.1 \cdot 10^{-3}$ (4.87) | $2.1 \cdot 10^{-3}$ (4.55) | $1.2 \cdot 10^{-3}$ (3.49) |
| | $\varepsilon\,(w)$ [ms$^{-1}$] | $7.1 \cdot 10^{-3}$ (1.06) | $1.8 \cdot 10^{-3}$ (1.12) | $1.1 \cdot 10^{-3}$ (0.71) |
| 3.2[a] | $\varepsilon_{min}\,(\Delta p)$ [Pa] | $4.3 \cdot 10^{-2}$ (21.8) | $4.0 \cdot 10^{-2}$ (102.5) | $2.7 \cdot 10^{-2}$ (6.1) |
| | $\varepsilon_{max}\,(\Delta p)$ [Pa] | $5.1 \cdot 10^{-1}$ (-193.6) | $8.8 \cdot 10^{-2}$ (-269.1) | $4.8 \cdot 10^{-2}$ (-41.6) |
| 3.3 | $\varepsilon\,(M_{flap})$ [Nm] | $2.9 \cdot 10^{-2}$ (36.6) | $4.5 \cdot 10^{-2}$ (56.5) | $3.4 \cdot 10^{-2}$ (42.9) |
| | $\varepsilon\,(M_{edge})$ [Nm] | $1.5 \cdot 10^{-2}$ (8.5) | $1.6 \cdot 10^{-2}$ (9.1) | $9.6 \cdot 10^{-3}$ (4.4) |

(a)   Minimum and maximum confidence interval of pressure taps

The values of the 95% confidence interval, see Table 6, are significantly smaller compared to those of the standard deviation (Table 5Table 5). The reason is the relatively large number of samples, $n \approx 3.6 \cdot 10^{3}$ in terms of the wake velocities, $u$ and $w$, and $n \approx 1.7 \cdot 10^{4}$ per azimuthal angle in the remaining cases. Hence, the presented average results are contained by a reasonably small confidence interval.

**Data availability.**

Measurement data and results can be provided by contacting the corresponding author.

**Author contribution**

Jörg Alber performed the wind tunnel experiments together with Rodrigo Soto-Valle and the support of all co-authors. Jörg Alber processed the data and prepared the manuscript with the support of Marinos Manolesos and Rodrigo Soto-Valle both of whom contributed with important comments and suggestions to all section of the manuscript.

**Competing interests**

The authors declare that they have no conflict of interest.

**Acknowledgements**

The authors would like to acknowledge the constant support of the BeRT project by the researches and the technicians of the Hermann-Föttinger Institut at the Technische Universität Berlin. The authors would also like to ~~acknowledge~~ appreciate the technical support of SMART BLADE ~~GmbH~~. Rodrigo Soto-Valle would like to thank the support of ANID PFCHA/Becas Chile-DAAD/2016-91645539. Marinos Manolesos would like to acknowledge the contribution of the EPSRC Supergen ORE Hub Early Career Researcher Research Fund.

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
