# Peer review of "Aerodynamic Effects of Gurney Flaps on the Rotor Blades of a Research Wind Turbine"

_Wind Energy Science, 2020_

## Referee Comment (RC1) · Anonymous Referee #1 · 2 Mar 2020

Dear authors, your study is clear and well presented. The topic (use of GFs as a retrofit for HAWTs) is of interest. The peculiar characteristics of your case study - although honestly discussed - limit a little bit the impact and the generality of the results. Overall, I would recommend it for publication, provided that some corrections are made. In particular:

1) You often use the verb "provoke", but very often not with the proper meaning. I would suggest replacing it in several instances. Overall, a revision by a native English speaker is suggested.

2) It would be useful to have a quantification of the experimental errors. Please also add error bars in Figure 11

3) Figure 9 (and the corresponding ones in the appendix) are not very redable. Please made lines thicker and/or manage the axes scale

4) The "1/2" in Eq. (9) is quite unusual. This formulation, however, is not coherent with the expression of Eq. 11. Please discuss and/or correct

5) Please expand the comments about the blockage effects. Beyond the aggregate BF, do you believe that the massive blockage could induce spanwise variation of the AoA? In other words, could BF alter the relative effect of GF depending on the span location?

6) To add some impact to the work, it would be nice to re-calculate the AoA by simulating the airfoil with CFD and to try comparing the pressure distribution with the experimental one. Do you think this could be feasible?

---

## Author Response (AR1)

**WES-2020-40: Aerodynamic Effects of Gurney Flaps on the Rotor Blades of a Research Wind Turbine**

Dear Reviewers and Editors

the following pdf-document consists of the following parts:

**Part 1:** Detailed answers to RC1

**Part 2:** Detailed answers to RC2

**Part3:** Marked-up changes in the manuscript:

Apart from minor changes, it is specified in the commentary field (to the right) to which of the reviewers the changes refer to. All mark-ups are included, so that the document looks a bit chaotic. The identical (new) manuscript, without mark-ups has been uploaded in the corresponding menu.

Thank you for your time and cooperation,

Please contact me in case there is anything missing,

best regards,

Jörg Alber

**Part 1**

**Reply to Anonymous Referee #1**

Dear Referee,

Thank you for the review and the helpful comments. They will all be included into the final version of the paper. The changes will improve the quality of the paper. In the following, I am responding to each of your remarks.

> *1) You often use the verb "provoke", but very often not with the proper meaning. I would suggest replacing it in several instances. Overall, a revision by a native English speaker is suggested.*

Reply: I cannot find the term "provoke" anywhere in the text. Yes, the final version will be corrected by a native speaker in order to improve the overall readability.

> *2) It would be useful to have a quantification of the experimental errors. Please also add error bars in Figure 11*

Reply: Agreed. The uncertainty quantifications will be included in the final version. In order to assure readability, the estimation of the experimental error as well as the subsequent error-propagation, might be added in the form of separate figures or tables.

Figure 11 is based on the different experimental results of the Clark-Y airfoil. No uncertainty estimation is provided in the external document (Kheir-Aldeen, 1996).

> *3) Figure 9 (and the corresponding ones in the appendix) are not very redable. Please made lines thicker and/or manage the axes scale*

Reply: Agreed. The figures will be adjusted.

> *4) The "1/2" in Eq. (9) is quite unusual. This formulation, however, is not coherent with the expression of Eq. 11. Please discuss and/or correct*

Reply: The definition of the axial induction directly inside the rotor wake would not include the "½" in the equation. However, based on the decelerated **wake-flow** downstream, the axial induction is defined according to Burton (2nd edition), p. 42:

$$U_W = (1 - 2a)U_\infty \qquad (3.8)$$

That is, half the axial speed loss in the stream-tube takes place upstream of the actuator disc and half downstream.

Hence, Eq.9 is coherent with Eq.11, i.e. calculating the AoA based on the induction factors, the undisturbed inflow-speed and the rotational speed. I will clarify the definition of Eq.9 in the text.

*5) Please expand the comments about the blockage effects. Beyond the aggregate BF, do you believe that the massive blockage could induce spanwise variation of the AoA? In other words, could BF alter the relative effect of GF depending on the span location?*

Reply: The relatively high blockage ratio is an inherent issue of the BeRT set-up. It has been investigated in previous studies based on both experiments (Bartholomay, 2017) and CFD simulations (Klein, 2018). Apart from the mentioned turbulence intensity, the axial velocities of both the inflow and the wake are considered homogeneous inside the rotor area. Hence, no significant spanwise AoA-variation could be detected for different measurement methods (3-hole

probes, Ultrasonic Anemometers, CFD) at least in the mid-span region for 0.45R < r < 0.85R. Nonetheless, the blockage effects are more pronounced close to the tip due to the strong flow-acceleration between the tip and the wind tunnel walls.

For the purpose of this study, the AoA are only determined at a local span-wise position of r = 0.56R by means of an Ultrasonic Anemometer, comparing the baseline to the GF configurations. The spanwise blockage effects will be discussed in more depth in the final version.

> *6) To add some impact to the work, it would be nice to re-calculate the AoA by simulating the airfoil with CFD and to try comparing the pressure distribution with the experimental one. Do you think this could be feasible?*

Reply: Previously, the BeRT set-up has been extensively investigated via URANS simulations, as published by Klein et al. (2018): https://www.wind-energ-sci.net/3/439/2018/.

This includes both the axial wake-velocity and the local AoA (Fig.16, Klein et. al) in the mid-span blade region, which are in agreement with the experimental results of this study. However, the cp-distribution has not been included in the mentioned paper of Klein et al..

A renewed CFD simulation of the BeRT rotor or the Clark-Y airfoil is beyond the scope of this study.

**Part 2**

**Reply to Referee #2**

Dear Athanasios Barlas (Referee),

Thank you for the detailed review and the helpful comments.

All your proposed changes, especially regarding the conclusions, will be included into the final version. They will improve the quality of the paper.

In the attached document, I am responding to each of your remarks in detail.

On behalf of all authors,

best regards,

Jörg Alber

**Direct answers to comments**

Page 1 (L28)
Isn't the effect on power extraction more important than fatigue loads in the inner region? The contribution of the inner sections to both power and fatigue loads is small, but probably the former is most important.

Answer:
I think it is both. There are studies on the effects of VGs in the inner region, contributing to both the alleviation of dynamic loads by stall-delay and, subsequently, to a power increase of 1-3%. But you are right, I will clarify this point in the introduction, plus adding a more specific reference.

Page 1 (L29):
This is probably more important for outboard blade sections.

Answer:
Yes, the effect of LE-roughness (LER) is more relevant outboard. Hence, the motivation of using GFs is two-fold: stall-delay in the inner region and compensation of LER in the outer parts. Both are likely to be leading to a power increase. I will also clarify this point in the motivation.

Page 3 (L45)
A reference would be necessary for such a statement.

Answer:
Agreed. The available reference will be included.

Page 5 (L104):
It is not clear if all measurements are corrected to account for the blockage effects (in order to draw conclusions for a free flow scenario) or not.
Answer:
No blockage correction is applied. All results refer solely to the 'reference' inflow conditions which are measured by means of two parallel Prandtl tubes. I will clarify this point in the final version.

Page 6 (L130):
It is not clear what this actually means.

Answer:
This statement can be ignored as it is not part of the (final) version available at the WESC discussion forum.

Page 6 (L132):

Please comment on the influence of the low test Reynolds number on the results, compared to a full scale case.

1.) The Re numbers of this table are based on the experiments (pressure tabs) and NOT the BEM simulations. This had already been changed in the last version of the paper.  2.) Inevitably, the low Re number reduces the scalability of the results, particularly compared to big machines with Re of several million.  Nonetheless, the effect of GFs mainly depends on the ratio between the GF-height and the BL thickness. As such, the results on the overall impact of GFs (lift&drag increase) on the Clark-Y profile at Re=250k is in agreement with airfoil experiments at higher Re numbers (1M to 2M) considering the much thinner BL-thicknesses.    I will discuss this approach in more depth in the final version.

Page 8 (L172):
Is the number of taps good enough for reasonable accuracy in the integrated forces? Any prior study?

Answer:
We are using a total of 29 pressure tabs at r=0.45R. For comparison: The NREL Phase VI research turbine uses 22 pressure tabs at each span-wise position (Hand (2001, p.30)) and the MEXICO rotor 29 tabs at each span-wise position (Snel (2009).  Furthermore, the cp-curves are in agreement with XFOIL simulations of the 2D-Clark-Y airfoil, as shown bv Soto-Valle (2020).  I will specify this information in the final version.

Page 9 (L200):
Is there any correction applied to the inflow speed, in order to derive the local angle of attack? The measurement position is quite far from the rotor plane (1.3R).

No (blockage-) correction is applied to the inflow speed.  In this study, the AoA are determined by relating the axial and tangential wake-velocities to the axial inflow-velocities. As such, 1.3R downstream is (just about) sufficiently far-away from the rotor plane. The paper of the MEXICO rotor serves as a reference for this statement (Snel (2009)).

Page 11 (L246):
This effect could be substantial considering that the rotor does not operate in optimal conditions. Please comment.

Answer:
The fact that the rotor is operating in sub-optimal conditions is discussed at different parts of the text. The rel. high blockage is most visible in terms of the high axial wake-velocity and, subsequently, in the form of rel. high AoA (AoA=9° versus AoA=5°), previously investigated by Klein (2018). Yes, this effect is relevant as it affects the conclusions regarding the flow-conditions in an "usual" far-field situation. However, the main aerodynamic impact of GF on the rotor performance is considered plausible and valuable, despite the wind tunnel effects  (in short: decrease of axial wake-velocity & AoA and increase of lift and bending moments).

Page 12 (L271):
To which radial position on the blade does this correspond to?

Answer:
The radial position is specified just above the figure: 0.56 R. I will include the radial position in the description of the Figure, too.

Page 14 (L292):
Maybe the no GF case should also be included in the plot? Maybe combine the first two plots?

Answer:
Combining all cases makes the plot very hard to read. The third plot brings the two separate plots back together by clarifying the changes between baseline versus GF configuration.

Page 14 (L292):
Again, the radial station these calculations correspond to should be clearly mentioned here.

Answer:
Agreed.

Page 15 (L314):

Any characterization of impact of GF on Cl/Cd including viscous drag would be interesting, although no measurements are available.

Answer:
Unfortunately, there are no adequate results that could be added.

Page 16 (332):
Is it only rotor vibrations or other inflow effects (e.g. flow misalignment) ? Is there any nP component?

Answer:
This statement is not part of the latest version as it is beyond the scope of this investigation. The structural-dynamic excitation is primarily a mechanical problem in the set-up (e.g. slight misalignement of bearings and main shaft) as previously investigated by Bartholomay (2017).

Page 16 (L346):
Comment for all plots: Any uncertainty quantification would be valuable, since changes are small.

Answer:
Agreed. The uncertainty quantification will be included in the final version. In order to assure readability, the estimation of the experimental error, as well as the subsequent error-propagation, might be added in the form of separate figures or tables.

Page 17 (L367):
It would be preferred if the conclusions are solely reflecting the analysis of the measurements, since it is not trivial to draw conclusions on general effects on a full scale wind turbine.

Answer:
Agreed. The conclusion will be changed accordingly.

[revised manuscript text omitted]

**Kommentiert [D10]:** RC2, Line 130f (Table 1) (clarification on small Re numbers)
RC2, Line 130: No changes have been made regarding the comment on the BEM method, see direct reply to RC2.

[revised manuscript text omitted]

Kommentiert [D11]: RC2, Line 173f: The NREL test turbine serves as a reference that the amount of pressure taps is reasonable.

The required AoA, α, are adopted by the uncorrected inflow and wake velocity measurements  (Sect. 3.1). the term $c_t \cdot \sin(\alpha)$ in Eq. (5) solely describes the pressure drag, which does not contain  the skin-friction drag, so that $c_t \cdot \sin(\alpha) < c_d$. Moreover, for relatively small AoA, $c_t \ll c_d$ (Barlow, 1999).

**Kommentiert [D12]:** RC2, Line 200, (clarification inflow speed correction)

**2.3.3 Strain gauges**

The Strain Gauges (SGs) are mounted at the clamping of the blade  (Figure 6 (a)) detecting the Root Bending Moments (RBMs) in the out-of-plane or flapwise and in-plane or edgewise direction. They are connected in a full-bridge configuration aiming at the mitigation of temperature and cross talk effects (FAET-A6194N-35). The experimental procedure to determine the RBMs is based on Bartholomay et al. (2018). For the purpose of the  presented baseline measurements, a simplified post-processing protocol is applied without including the data-based cross talk correction.

Before testing each blade configuration, the offset signal is recorded in slow-motion at the lowest rotating frequency available, $f_{rot} = 0.1$ Hz. In this way, the gravitational RMBs are subtracted from the results, which are otherwise registered as a sinusoidal signal in the edgewise direction. At operational frequencies, the axial forces due to the blade rotation are causing a material deformation directed towards the blade tip. They are quantified as a combination of centrifugal and gravitational forces by

$$F_{axial} = F_{cent} - F_{grav} = (m_{blade} \cdot r_{cg} \cdot \omega^2) - (m_{blade} \cdot g \cdot \cos(\varphi)), \tag{6}$$

where $m_{blade} = 5.67$ kg, the center of gravity is located at $r_{cg} = 0.31R$, $g$ is the gravitational constant and $\varphi$ refers to each phase-locked blade position. The rotational frequency, ω, is kept constant during each test-run so that the centrifugal force $F_{cent}$ becomes a constant correction term at each OP. The effective flapwise and edgewise RBMs, which are related exclusively to the aerodynamic loads acting on the blade, are  determined by

$$M_{flap}(\varphi) = (U_{f,\mathrm{raw}}(\varphi) - U_{f,off}(\varphi)) \cdot K_{f1} - (F_{axial} \cdot K_{f2}), \tag{7}$$

and

$$M_{edge}(\varphi) = (U_{e,\mathrm{raw}}(\varphi) - U_{e,off}(\varphi)) \cdot K_{e1} - (F_{axial} \cdot K_{e2}), \tag{8}$$

where

- $M_{flap}$ and $M_{edge}$ are the aerodynamic flapwise or edgewise RBMs in Nm.
- $U_{f,raw}$ and $U_{e,raw}$ stand for the raw data signal in V.
- $U_{f,off}$ and $U_{e,off}$ describe the slow-motion offset signal in V.
- $K_{f1}$ and $K_{e1}$ refer to constant calibration factors to transform V into Nm.
- $K_{f2}$ and $K_{e2}$ refer to constant calibration factors to transform the axial forces from N into Nm.

Applying Eq. (7)(7) and (8) both the out-of-plane and the in-plane RBMs are computed for each of the 36 blade positions, as shown in Sect. 3, as follows..

**3 Results**

[revised manuscript text omitted]

(a)                                    (b)                                    (c)

[Figure]

**Figure 8.** Angles-of-attack in the tripped case at $r = 0.56R$ and $\varphi = 270°$. (a) Stall and rated conditions (b) Rated and feather conditions (c) AoA difference between Gurney flap configuration and the baseline.

At rated conditions, the AoA of the baseline case amount tois $\alpha_{ZZ} = 8.8°$, see Figure 8Figure 8 (a) and (b). This outcome is in agreement with comparable previous investigations in the mid-span region based on 3-hole probes as well as URANS CFD simulations of the BeRT, as detailed by Klein et al. (2018). Hence, the AoA are considered stable with respect to the mid-span region, i.e. $0.65R < r < 0.45R$. Furthermore, Figure 8Figure 8 (c) displays the consistent AoA decrease caused by the GF configurations. Depending on the GF-height, it amounts to $\Delta\alpha_{GF=0.5\%c} = 0.5°$ and $\Delta\alpha_{GF=1.0\%c} = 0.9°$, i.e. to a more favorable level in terms of the BeRT rotor. Hence, As such, thethe results quantify a crucial of the crucial effects of retrofitted GFs on the blade performance; decreasing axial wake velocities and thus reduced AoA.

In the following Sect. 3.2, the changing AoA are correlated to the local lift normal force coefficients, $c_n$, in order to obtain in the mid-span blade regionthe lift coefficients, $c_l$.

**3.2 Pressure distribution and lift performance**

Figure 9 shows the distribution of the pressure coefficients, $c_p$, for regarding the different OPs.

[Figure]

(a)            (b)            (c)

Normalized chord length

—○—Baseline(ZZ)   ····GF=0.5%c   — –GF=1%c

**Kommentiert [D19]:** RC 1, No. 3 (manage scales and improve readability of plots)

[revised manuscript text omitted]

**Kommentiert [D24]:** RC2, Line 367ff (keep conclusions to experimental results)

 **Appendix A: Results of the clean case**

[Figure]

**Figure A 1.** Clean case at $r = 0.56R$ and $\varphi = 270°$. (a) Axial and tangential wake veloc(ies) normalized by the inflow velocity. (b) Standard

[Figure]

deviation of the wake veloc(ities) normalized by the average wake veloc(ities).

[Figure]

**Figure A 2.** Angles-of-attack in the clean case at *r* = 0.56R and φ = 270°. (a) Stall and rated conditions (b) Rated and feather conditions (c) AoA difference between Gurney flap configuration and the baseline.

[Figure]

Normalized chord length

—○—Baseline(clean)   ····GF=0.5%c   – –GF=1%c

430 **Figure A 3.** Pressure distribution in the clean case with respect to different scales at $r = 0.45R$ and $\varphi = 270°$. (a) $TSR = 3.0$. (b) $TSR = 4.3$. (c) $TSR = 5.6$.

[Figure]

**Figure A 4.** Flapwise and edgewise root bending moments in the clean case. (a) $TSR = 3.0$. (b) $TSR = 4.3$. (c) $TSR = 5.6$.

**Appendix B: Uncertainty estimation**

**Kommentiert [D25]:** This Appendix is new. It refers to
-RC1, No. 2 (add uncertainty estimation)

[revised manuscript text omitted]

---

## Referee Report (RR1)

The manuscript presents an experimental study on a 3 m diameter horizontal wind turbine rotor. The focus was on the effects of Gurney flaps on the aerodynamic performance of the blade. Two Gurney flaps heights were tested on a clean blade and with turbulator tape. Experiments studies are always welcome, whether for the knowledge provided and/or for data generated that can be used to validate numerical models.

Only few attempts to physically comment the experimental observations reported. This should be improved in the revised version of the manuscript.

My comments and suggestions are presented as follows:

1. A linguistic revision of the manuscript is recommended.
2. The introduction needs to be improved to show the relevance of the presented experiment. Can author briefly mention what are the wide knowledge gaps and what portion of that current study is trying to fill. List point-by-point objectives and map the achievements of objectives in conclusion.
3. Line 38: add (c) after chord-length.
4. Line 48: correct the range (1.3 %c < *GF* <3.5 %c).
5. Line 87: the authors stated a blockage ratio of approximately 40 %. Could authors elaborate on this (how did they evaluate this value; what impact will have this blockage on the results...)
6. Line 109: mention that XFOIL is a program.
7. Section 2.2.1-2.2.2: To adjust the dimensions of the ZZ tape and the GF, the authors used XFOIL to evaluate the BL thickness. XFOIL is a 2D airfoil calculator, could authors explain how can exploit this result to a blade flow which is 3D?
8. Please rearrange Table 1 for clarity. Tripped case is missed. Measurement method part could be arranged in another table.
9. Lines 138-139: the authors stated that their findings are relevant beyond the Re-numbers of the BeRT blades, as long as the GF/BL ratio is kept constant. Are you suggesting that 2 flows with different Re numbers are comparable? Please explain more about this statement and define the ratio (GF/BL) in the text.
10. Line 217: RMBs#RBMs
11. Curves in graphs with 2 vertical axes must be correctly identified. For example, in figure 7a, indicate which curves correspond to axial and tangential velocities.
12. Line 245: free flow conditions may be confusing (flow without turbine or without wind tunnel walls).
13. Lines 245-253: The authors showed that the axial wake velocity is more sensitive to the wind tunnel blockage than the tangential velocity, could you please explain this phenomenon?
14. Line 276: Define $\Delta\alpha_{GF}$
15. Line 277: Please what do you mean by '' *to a more favourable level in terms of the BeRT rotor* ''? I think it is better to reformulate lines 277-278.
16. Is there a reason to evaluate the angle of attack at r=0.56R and the pressure coefficient which depends on this angle at a different radial position (r=0.45R)?
17. Line 285: Define $\Delta Cp$.

18. Lines 285-290: Description of Figure 10 needs to be rewritten more clearly to show the effect of GF on the pressure coefficient.
19. Line 312: change $\Delta cl$ to cl.
20. Regarding the conclusion, please see comment #2.

---

## Author Response (AR2)

Dear Referees,

First of all, I'd like to thank you for your time and providing helpful comments that have improved the quality of this manuscript. In the following, I am responding point-by-point to each of your remarks in green letters. The corresponding changes were included with mark-ups in the final version of the manuscript (see below).

On behalf of the authors,

best regards,

Jörg Alber

**Point-by-point reply to Referee #1 (RC1, Galih Bangga)**
Received: 10th of August 2020

Dear Authors,

The investigated studies are of interest for wind turbine community and have been conducted thoroughly. After evaluating your paper and previous remarks from the reviewers, I agree with most of your answers but further detail shall be given concerning the angle of attack. Below please find some minor critics that I wish to be considered before the paper can be published:

1. If I follow your paper correctly and by reading your answers to one of the reviewers (including the paper from Annette), the angle of attack was obtained in the measurement by assuming 2D flow. This was done by comparing the rotating rotor with a 2D polar data. This relatively simple approach should work well if no flow separation or unsteady effect presents. However, as you introduce the Gurney flaps, there will be little unsteadiness occurring near the trailing edge of the blade. How did you take this into account?

> In this paper, no 2D airfoil assumptions or simulations have been used for any of the AoA calculations. The AoA is obtained experimentally by means of the axial and tangential wake velocities (Sect. 3.1). The only 2D flow assumptions were used to estimate the boundary layer thickness in order to determine the height of both the ZZ-tape and the GFs (with XFOIL).

> Previously, Annette Klein et. al. published experiments on the BeRT using a 3-hole probe set-up (p.446) in the mid-span region (r=0.65R). For that, 2D polar data (cl) was used in order to calculate the AoA. These previous investigations are used as a reference to the tests that are published in this new paper.

> Considering the question at the end: there are numerous publications showing that Gurney Flaps (GFs) lead to increased unsteadiness near the trailing edge. This is the main reason for drag increase. However, the more the GF is submerged "deep enough" into the boundary layer (i.e. the smaller it is), the less pronounced is the unsteadiness that appears (see Fig. 2). Regarding the unsteadiness caused by the GF configurations that were tested in this campaign, the 3D Ultrasonic measurements in the wake do not reveal increased turbulence levels, see Fig. 7 (b).

2. The source of the 2D polar data for the AoA calculation is also unclear. Was the Gurney flap also included here?

> No 2D polar data has been used for the calculations of the presented results. The only (external) 2D lift polars are presented as a reference to our results, see Fig. 12.

3. Furthermore, the analysis was done mostly at 0.45R, and you claimed that the AoA evaluation has been confirmed with FLOWer calculations in the other paper. Despite that, the comparison done in Annette's

paper was at the spanwise positions no less than 0.65R. How did you ensure the 2D assumption will be correct especially for the lower TSR case? I do agree that lift will not be affected significantly for little inaccuracy in AoA calculation in the mid-span region, however drag will still be influenced as I have seen for example on the AVATAR rotor (see Fig. 7 page 13 https://aip.scitation.org/doi/pdf/10.1063/1.4978681).

> The blade positions with respect to the measurement methods were clarified in the revised version of the manuscript, see the new Table 3.
>
> The assumption of constant AoA throughout the mid-span region ($0.45R < r < 0.65R$ ) is explained in more depth (see new Table 4)
>
> However, it is not possible to measure the total drag coefficients ($c_d$) of any blade-element of the BeRT, see following comment no. 4 and 5.

4. It is interesting to note that you state the drag penalty is still at reasonable value (line 115) but this has never been proven in the paper.

> This statement is a generally accepted design-consideration by numerous airfoil experiments and/or CFD simulations: the airfoil drag, $c_d$, increases in relation to the GF height. Vice versa, the more the GF is submerged by the BL layer, the smaller the drag-increase becomes (see Sect. 1).
>
> This statement was clarified in the new version of the manuscript. In addition, the experimental results of the pressure drag were included, see following comment 5.

5. Lastly, I would recommend to add a plot showing the lift over drag to really assess the performance of the rotor. The drag might be estimated by the pressure integration, although friction drag is also important for small AoAs.

> The tangential force coefficient, $c_t$, can be transformed into the pressure-drag component, $c_{dp}$, which does not include the skin-friction drag component (see Sect. 2.3.2). Hence, presenting only the values for pressure-drag is, by default, incomplete information.
>
> As suggested, the plots of the pressure drag ($c_l$ over $c_{d,p}$) plus explanations were included as an indicator for the drag performance of the blade element, with and without the use of GFs, see Sect. 3.2.

Small typos
Line 275 i.e. $0.65R < r < 0.45R$ -- > should be other way around.

> Done.

Line 13 Reynold --> Reynolds. It is a person name and please check for some others….

> Done.

God luck!

> Thank you!

Kind regards,
Galih Bangga

**Point-by-point reply to Anonymous Referee #2 (RC2)**
Received 10th of August 2020

The manuscript presents an experimental study on a 3 m diameter horizontal wind turbine rotor. The focus was on the effects of Gurney flaps on the aerodynamic performance of the blade. Two Gurney flaps heights were tested on a clean blade and with turbulator tape.
Experiments studies are always welcome, whether for the knowledge provided and/or for data generated that can be used to validate numerical models.
Only few attempts to physically comment the experimental observations reported. This should be improved in the revised version of the manuscript.

My comments and suggestions are presented as follows:

1. A linguistic revision of the manuscript is recommended.

   Done.

2. The introduction needs to be improved to show the relevance of the presented experiment. Can author briefly mention what are the wide knowledge gaps and what portion of that current study is trying to fill. List point-by-point objectives and map the achievements of objectives in conclusion.

   The objectives and the relevance of the study were specified in both introduction and conclusions.

3. Line 38: add (c) after chord-length.

   Done.

4. Line 48: correct the range (1.3 %c < $GF$ <3.5 %c).

   Done.

5. Line 87: the authors stated a blockage ratio of approximately 40 %. Could authors elaborate on this (how did they evaluate this value; what impact will have this blockage on the results...)

   The definition of the blockage ratio was specified. Otherwise a reference to Sect. 3.1 (results) was included, where the effects of blockage are discussed in more depth.

   (In general, the most tangible effect of the wind tunnel blockage is the fact that the axial wake velocity is much higher than expected. As a direct consequence, the AoA are elevated, too.)

6. Line 109: mention that XFOIL is a program.

   Done.

7. Section 2.2.1-2.2.2: To adjust the dimensions of the ZZ tape and the GF, the authors used XFOIL to evaluate the BL thickness. XFOIL is a 2D airfoil calculator, could authors explain how can exploit this result to a blade flow which is 3D?

   The sections were rewritten clarifying that the flow is attached (pre-stall) and thus assumed 2D in order to estimate the BL-thickness by means of the XFOIL code.

8. Please rearrange Table 1 for clarity. Tripped case is missed. Measurement method part could be arranged in another table.

    The tables were split up and rearranged for more clarity, as suggested: see new Table 1, 2 and 3.

9. Lines 138-139: the authors stated that their findings are relevant beyond the Re-numbers of the BeRT blades, as long as the GF/BL ratio is kept constant. Are you suggesting that 2 flows with different Re numbers are comparable? Please explain more about this statement and define the ratio (GF/BL) in the text.

    This statement was shifted to Sect. 2.2.2 (dimensioning of GF-height), where it fits better than in combination with the test matrix. The design concept of the GF/BL-ratio was explained in depth and with more clarity.

10. Line 217: RMBs#RBMs

    Done.

11. Curves in graphs with 2 vertical axes must be correctly identified. For example, in figure 7a, indicate which curves correspond to axial and tangential velocities.

    Done.

12. Line 245: free flow conditions may be confusing (flow without turbine or without wind tunnel walls).

    Done.

13. Lines 245-253: The authors showed that the axial wake velocity is more sensitive to the wind tunnel blockage than the tangential velocity, could you please explain this phenomenon?

    The tangential velocity-component depends primarily on the rotational speed of the blade. Hence, it is less affected by the rapid axial velocity component. This observation was commented on in more depth.

14. Line 276: Define ΔαGF

    Done.

15. Line 277: Please what do you mean by '' *to a more favourable level in terms of the BeRT rotor* ''? I think it is better to reformulate lines 277-278.

    The statement was reformulated in order to make this point clearer.

    (The reason why the BeRT performance is improved by the GFs is because it is running sub-optimally due to the wind tunnel blockage effects and/or ZZ tape. Otherwise, if BeRT ran at its optimum, the impact of GFs would probably have an adverse effect on the performance)

16. Is there a reason to evaluate the angle of attack at r=0.56R and the pressure coefficient which depends on this angle at a different radial position (r=0.45R)?

The (practical) reason is that the AoA were evaluated by different experiments at different span-wise positions along the mid-span area:

- 0.45R -> pressure tabs (reference to Soto-Valle et al.)
- 0.56R -> Ultrasonic Anemometer
- 0.65R -> 3-hole probes (reference to Klein, et. al).

The new table 4 summarizes the different research efforts with more clarity.

Based on the mentioned results, the AoA are assumed constant in the mid-span area for 0.65R < r < 0.45 (due to the built-in twist angles of the blades). This assumption was explained in a clearer way in the new version of the manuscript.

17. Line 285: Define ΔCp.

    Done.

18. Lines 285-290: Description of Figure 10 needs to be rewritten more clearly to show the effect of GF on the pressure coefficient.

    Done. The paragraph in question was re-written.

19. Line 312: change Δcl to cl.

    Actually, the purpose of Fig. 12 is the comparison between both cl and Δcl, i.e. the cl of the baseline and the shift of Δcl when implementing a GF. The statement was rephrased for clarification.

20. Regarding the conclusion, please see comment #2.

    The conclusions were re-written in accordance with the suggested changes from comment #2.

[revised manuscript text omitted]

Kommentiert [D13]: RC2, #.8: rearrange tables

[revised manuscript text omitted]

(a) (b) (c)

Pressure coefficient $c_p$

Normalized chord length

—○—Baseline(ZZ)    ·····GF=0.5%c    – –GF=1%c

**Figure 10.** Pressure  coefficients in the tripped case with respect to different scales at $r = 0.45R$ and $\varphi = 270°$. (a) *TSR* = 3.0. (b)
350 *TSR* = 4.3. (c) *TSR* = 5.6.

The $c_p$ curves shown in Figure 10 (b) and (c) represent the pre-stall cases at $\alpha_{TSR=4.3} = 8.8°$ and $\alpha_{TSR=5.6} = 4.8°$, respectively. At stall,  see Figure 10 (a), the separation at the SuS is not yet complete despite the elevated AoA, $\alpha_{TSR=3.0} = 16.3°$.  indicat the effect of stall delay due to the blade
355 rotation, as discussed hereafter.

Moreover, the GF configurations cause an expansion of the pressure differences between the PrS and the SuS, $\Delta c_p$, along the complete chord-length and regarding all OPs. This effect is particularly visible in terms of the aft-loading towards the TE at $0.5 < x < 0.9$. As such, $\Delta c_p$ reflects the increased circulation due to the GF applications, as reported by Storms and Jang (1994)
360 based on the clean NACA 4412 airfoil ($th_{max} = 12.0$ %c, Re = $2.0 \cdot 10^6$).

In order to quantify the results, the $c_p$ distribution is transformed into the local lift curve based on Eq. (5(5). The required AoA are adopted from Sect. 3.1, so that the lift coefficients combine the results of both the wake velocity and the pressure measurements.

365

[Figure]

[Figure]

(a)

(b)

(c)

[Figure]

**Figure 11.** Lift coefficients over angles-of-attack at $r = 0.45R$ and $\varphi = 270°$. (a) Tripped case. (b) Clean case. (c) Relative lift increase of Gurney flap configurations in relation to the corresponding baseline.

Figure 11Figure 11 (a) and (b) depict the lift coefficients of both the tripped and the clean cases. Starting from the baseline, the tripped case shows smaller $c_l$ at $4° < \alpha < 5°$ because of the forced BL transition at the LE. At $8° < \alpha < 9°$, this is not the case anymore, while in the stall region, $15° < \alpha < 17°$, the ZZ tape appears to develop a beneficial effect on the lift performance. This phenomenon is probably caused by the tripped and more turbulent BL that remains attached until closer to the TE. In the clean case, however, the less energetic BL separates earlier thus leading to smaller $c_l$ at elevated AoA. This observation is confirmed by comparable airfoil experiments on the FX 63-137 airfoil section at ($th_{max} = 13.7 \%c$, Re = 1.0·10⁵ < $Re$ < 2.0·10⁵) using ZZ tape with a thickness of 0.75 mm (Holst et al., 2016). Despite the decrease in the pre-stall, the lift coefficients are found on a similar level in the post-stall region.

Furthermore, lLooking at the GF configurations (Figure 11 (a) and (b)), the lift $c_l$ performance in the tripped case is on a similar, or even higher level considering the complete AoA range, 4° < α < 17°. Hence, forced LE transition is does not neutralize or mitigate mitigating or neutralizing the GF effect. In fact, the GF configurations appear to alleviate the adverse effects of LE roughness by improving the local lift $c_l$ performance. Figure 11Figure 11 (c) summarizes highlights the relative $c_l$ lift increase, $\Delta c_l$, of bothbetween the GF configurations in relation toand the corresponding baseline configurationseases. At rated conditions, $TSR$ = 4.3, $\Delta c_{l,GF=0.5\%c}$ = 0.11 or 9.3 % and $\Delta c_{l,GF=1.0\%c}$ = 0.19 or 16.9 %, illustrating the main characteristic of retrofit GFs; the considerable lift increase.

Moreover, Tthe scale level of both $c_l$ and $\Delta c_{l,GF=1.0\%c}$ is in agreement with comparable wind tunnel experiments based on a similar Clark-Y airfoil section, as depicted in Figure 12Figure 12.

[Figure]

**Figure 12.** Lift coefficients of a the Clark-Y airfoil including Gurney flap, reproduced and modified from Kheir-Aldeen (2014).

[revised manuscript text omitted]